# Protocol for a multicentre prospective exploratory mixed-methods study investigating the modifiable psychosocial variables influencing access to and outcomes after kidney transplantation in children and young people in the UK

Ji Soo Kim [1,2] Jo Wray [3] Deborah Ridout,[4] Lucy Plumb,[5,6] Dorothea Nitsch,[5,7] Matthew Robb,[8] Stephen D Marks [1,2]

For numbered affiliations see end of article.

**Correspondence to**
Dr Ji Soo Kim;
jisoo.kim@nhs.net

## ABSTRACT

**Introduction** Kidney transplantation is the preferred therapy for children with stage 5 chronic kidney disease (CKD-5). However, there is a wide variation in access to kidney transplantation across the UK for children. This study aims to explore the psychosocial factors that influence access to and outcomes after kidney transplantation in children in the UK using a mixed-methods prospective longitudinal design.

**Methods** Qualitative data will be collected through semistructured interviews with children affected by CKD-5, their carers and paediatric renal multidisciplinary team. Recruitment for interviews will continue till data saturation. These interviews will inform the choice of existing validated questionnaires, which will be distributed to a larger national cohort of children with pretransplant CKD-5 (n=180) and their carers. Follow-up questionnaires will be sent at protocolised time points regardless of whether they receive a kidney transplant or not. Coexisting health data from hospital, UK renal registry and National Health Service Blood and Transplant registry records will be mapped to each questionnaire time point. An integrative analysis of the mixed qualitative and quantitative data will define psychosocial aspects of care for potential intervention to improve transplant access.

**Analysis** Qualitative data will be analysed using thematic analysis. Quantitative data will be analysed using appropriate statistical methods to understand how these factors influence access to transplantation, as well as the distribution of psychosocial factors pretransplantation and post-transplantation.

**Ethics and dissemination** This study protocol has been reviewed by the National Institute for Health Research Academy and approved by the Wales Research Ethics Committee 4 (IRAS number 270493/ref: 20/WA/0285) and the Scotland A Research Ethics Committee (ref: 21/SS/0038). Results from this study will be disseminated across media platforms accessed by affected families, presented at conferences and published in peer-reviewed journals.

## STRENGTHS AND LIMITATIONS OF THIS STUDY

⇒ Prospective, longitudinal study design allows for a more detailed understanding of delays in transplantation associated with different psychosocial factors.
⇒ Combination of qualitative and quantitative data enables in-depth exploration of how psychosocial factors influence access to transplantation and outcomes thereafter.
⇒ Involvement of the majority of UK paediatric nephrology units ensuring representation of paediatric chronic kidney disease-5 population
⇒ Limited follow-up timeline will capture only short-term to medium-term rather than long-term outcomes.
⇒ Utilisation of interpreters in qualitative interviews increases involvement of non-English-speaking families; however, the lack of validated translations of questionnaires may limit involvement of non-English-speaking families in capturing quantitative data.

## INTRODUCTION

Around 1000 children (aged 0–17 years) with stage 5 chronic kidney disease (CKD-5) in the UK receive kidney replacement therapy (KRT) in the form of either peritoneal dialysis, haemodialysis or kidney transplantation.[1 2] Kidney transplantation is the gold-standard therapy for reducing mortality and improving outcomes for children.[3] Minimising time on dialysis in favour of transplantation has been shown to reduce CKD-5-related complications and morbidity.[4] Compared with dialysis, transplantation is also presumed to improve patients' health-related quality of life (HR-QoL).[5] Furthermore in the UK, for every year

that the patient's kidney transplant functions, transplantation is three times more cost-effective than dialysis for the National Health Service (NHS).[6]

However, not every child with CKD-5 can access kidney transplantation. There are approximately 193–217 prevalent children on dialysis each year.[7 8] Annually only 130–160 paediatric kidney transplants are performed in the UK.[9] There appears to be variation in transplantation access, practice and outcomes between UK paediatric nephrology units.[1 2 9] A cross-sectional survey conducted by the British Association for Paediatric Nephrology examined reasons for paediatric kidney transplantation delay in the UK. The survey showed that psychosocial factors make up 19% of the barriers, although specific factors were not identified in this study.[10] Compared with paediatrics, considerable research has been done in adult patients, regarding psychosocial factors implicated in transplant access. Formalised pretransplant and post-transplant psychosocial assessments have been widely researched for adult solid organ transplant recipients.[11–14] The UK-wide study, 'Access to Transplantation and Transplant Outcome Measures' specifically investigated psychosocial barriers in adult kidney transplant recipients.[15] Researchers found inequities in transplant access, in spite of a universal healthcare system, based on socioeconomic status, education level, health literacy and racial background.[16 17] In terms of outcomes, they found no difference in post-transplant HR-QoL between living and deceased donor recipients and that recipient expectations influenced post-transplant recovery.[15–19] However, for the UK children, it is less clear what these 'psychosocial factors' are and how they influence unit-specific decisions, access to kidney transplantation and outcomes.[10 20 21] We acknowledge the recent progress being made in exploring these psychosocial factors in some countries.[22–24] However, psychosocial studies for children with CKD-5 are still limited by retrospective or cross-sectional design or by single-centre or small study cohorts. There are no UK studies that prospectively explore how these factors impact paediatric kidney transplantation access over time.

## AIMS AND OBJECTIVES

This study aims to prospectively evaluate the psychosocial factors that are actual or perceived barriers to paediatric kidney transplantation which may be associated with poor transplant outcome. We anticipate these psychosocial factors to be broad at an individual, family and societal level and that they will encompass mental health and social determinants of health as defined by Marmot *et al*: 'the conditions in which people are born, grow, live, work and age'[25]—we hypothesise that psychosocial factors implicated in transplant access can be quantified using formal, validated measures and that these factors may influence outcomes in the short-term period following transplantation.

This will be achieved through the following research objectives:

1. Describe the psychosocial factors perceived by clinicians as barriers to kidney transplantation.
2. Describe current interventions (if any) implemented to address these psychosocial factors.
3. Explore the experiences and beliefs of children and their families regarding any psychosocial challenges or facilitators in accessing a kidney transplant.
4. Quantify the identified psychosocial factors implicated in CKD management.
5. Measure the prevalence of identified psychosocial factors (positive and negative) in the national cohort of patients being pre-emptively worked up for transplant, on dialysis or listed for transplant.
6. Examine these psychosocial factors regarding their association with time to transplant and their changes following transplantation.
7. Synthesise findings from each phase to inform recommendations about which psychosocial factors are potential barriers or facilitators to accessing kidney transplantation.

Findings synthesised from this study will inform the development of a complex intervention to improve uptake of kidney transplantation in children with CKD-5 who would benefit most from one.

## METHODS AND ANALYSIS
### Study design overview

This prospective study has three phases, commencing with a sequential exploratory mixed-methods design, followed by a sequential explanatory mixed-methods design (figure 1).[26] This approach was chosen to first gain new insights by becoming familiar with the range of potential psychosocial factors and then longitudinally observing the influence of these factors on kidney transplantation access and outcomes for children.[27] Phase 1 will consist of exploratory interviews with purposively selected participants. These interviews will inform which questionnaires will be distributed at baseline and follow-up to the wider cohort of children with CKD-5 and their carer(s) in phase 2 stage A. Using the interviews to inform questionnaire selection will ensure questionnaires are relevant to, and resonate with, families. Participant families with outlier findings from phase 2 stage A will then be invited for explanatory interviews in phase 2 stage B. Finally, in phase 3, the qualitative and quantitative data from the previous phases will be analysed together to develop an integrated understanding of psychosocial factors that influence access to kidney transplantation.

### Patient and public involvement

All elements of the study design were codeveloped and approved by our Research Partner Family and study steering group. Our Research Partner Family (a young person and parent dyad) and steering group members have lived experience of CKD, dialysis and kidney

**(QUAL → QUANT) EXPLORATORY DESIGN** ⟶ **(QUANT → QUAL) EXPLANATORY DESIGN**

**PHASE ONE**

**INTERVIEWS**
- Aim: Identify questionnaires for use in Phase Two
- Cohort: NHS professionals, CYP with CKD-5 awaiting kidney transplantation and their parents/carers

**PHASE TWO**

**QUESTIONNAIRES [STAGE A]**
- Aim: Observe longitudinal trajectory of psychosocial parameters pre- & post-transplant
- Cohort: UK cohort of CYP with CKD-5 awaiting kidney transplantation and their families

**INTERVIEWS [STAGE B]**
- Aim: Explore why some CYP have good or poor outcomes pre/post-transplant
- Cohort: Smaller cohort of CYP/parents/carers who answered questionnaires

**PHASE THREE**

**SYNTHESIS**
- Aim: Integrated analysis of Phase One and Phase Two findings to inform future practice
- Cohort: Steering Committee of Patient & Public Stakeholders

**Figure 1** Flow diagram of sequential mixed-method study design. CYP, children and young people; NHS, National Health Service.

transplantation in childhood either as a patient or carer. The Great Ormond Street Hospital Young Persons Advisory Group has also approved the research question and study design. The Young Persons Advisory Group consists of children affiliated with Great Ormond Street Hospital for Children NHS Foundation Trust as a patient or family member of a patient and therefore who have the lived experience needed to advise researchers designing studies involving children.

### Population
This study aims to include children, carer and staff participants from all 13 UK paediatric nephrology units, of which 10 units offer kidney transplantation surgery on site and 3 units which do not offer transplantation surgery at their own site but offer shared transplant care with nearby units.

In phase 1 and phase 2 stage A, children (aged 0–17 years inclusive) with CKD-5 on chronic dialysis, being worked up for pre-emptive kidney transplantation or on the waiting list for deceased donation (active or suspended) or awaiting living donation, will be invited to participate with their carer(s). Where appropriate, children aged 5 years and above will be invited to directly participate. For children aged under 5 years or those who are unable to assent or consent, their carers will be consulted and offered an opportunity to participate through proxy measures. Members of the multidisciplinary team (MDT) involved in pretransplantation workup at their local unit (eg, paediatric nephrologists, transplant surgeons, nurse

specialists, social workers, family therapists, play therapists and members of the psychosocial team) will be invited for interviews in phase 1 only. Patients or carer(s) who are unable to give informed consent or patients who have been deemed too unwell to participate or had a recent acute hospital admission in the last 14 days will not be approached to participate in the study.

### Phase 1 interviews
A purposive sampling matrix will be used to maximise the diversity of views. Criteria for sampling will include age, sex and ethnicity of children, modality of KRT, whether the kidney unit offers transplantation surgery or not and role in the MDT. Completed interviews will be analysed in parallel with ongoing recruitment. Participant recruitment will stop when no new themes are generated from the data.

Analysis of these interviews will contribute towards selecting questionnaires for use in phase 2 stage A.

### Phase 2 stage A questionnaires
In this phase, we will aim to recruit every child with CKD-5 who meets the inclusion criteria at pretransplant baseline with an anticipated recruitment rate of 60%–70%.

At the time of study design, the predicted annual numbers across all 13 UK paediatric nephrology units as per UK Renal Registry (UKRR) reports were as follows[1 2]:
- Number of dialysis patients on the transplant waiting-list in a year; n=70.
- Number of patients starting dialysis in a year; n=125.

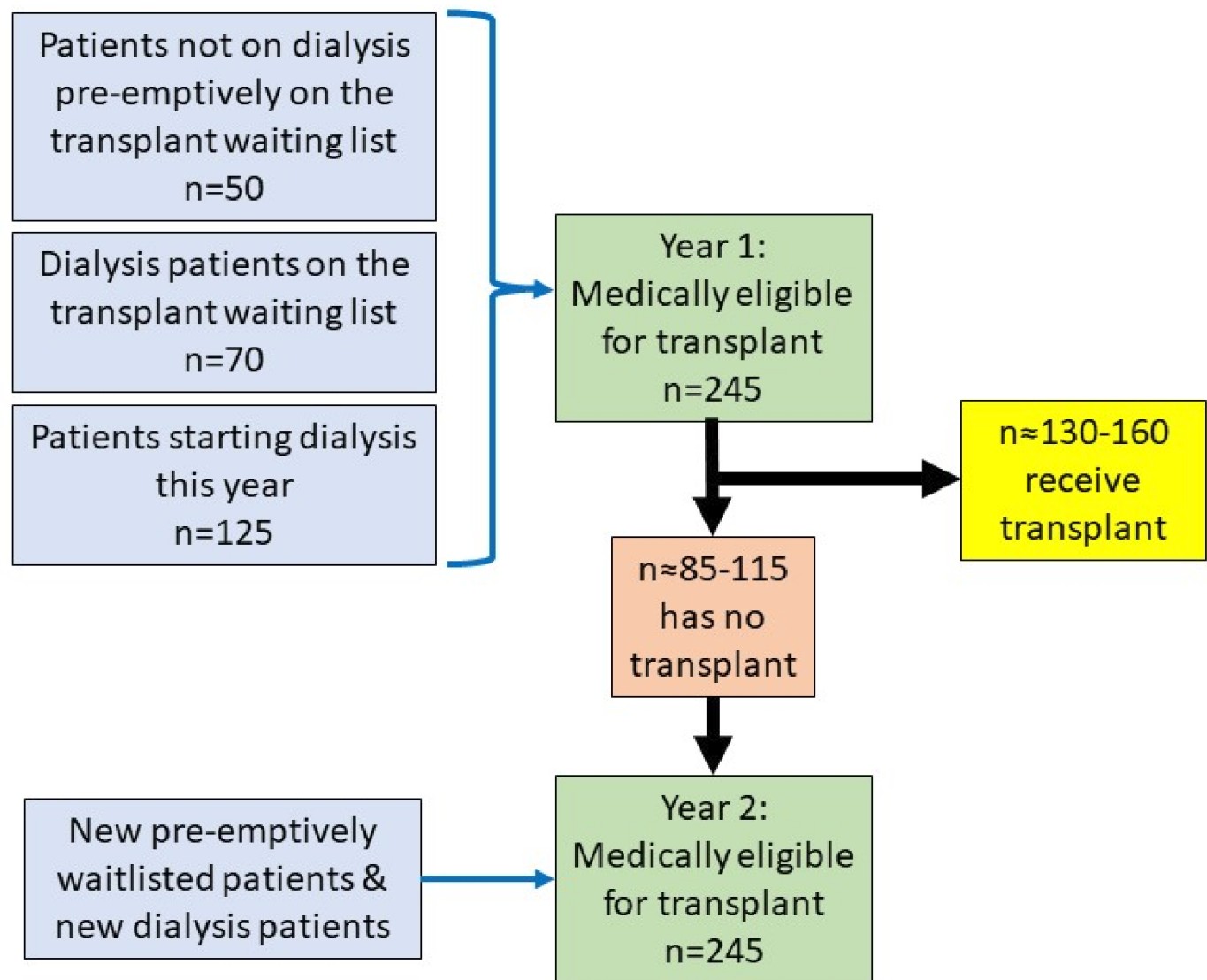

**Figure 2** Flow diagram illustrating predicted number of CYP who are eligible for transplantation and receive one across 2 years. CYP, children and young people.

► Number of patients not on dialysis but pre-emptively on the transplant waiting list; n=50.

► Transplant rate; 130–160 patients per year.

Based on these numbers, the predicted maximum number of eligible participants in a 1-year period would be 245. We chose a 2-year period for our sample size calculation, which would be a population size of 490, to include children who experience delays and receive their transplant after 1 year (see figure 2). We plan to recruit 180 patients and after allowing for a 20% drop-out, we expect to include 141 families. Assuming a prevalence of pertinent psychosocial factors in the national cohort of patients being pre-emptively worked up for transplant, on dialysis or listed for transplant of 15%, then with 141 families we will be able to estimate this with 5% precision. At the time of sample size calculation, there were no paediatric studies specifically measuring this nor studies encompassing all psychosocial factors. Therefore, we used the assumed prevalence of 15% based on a study

measuring psychological distress in potential adult transplant candidates.[28]

We expect some overlap in the cohort between phase 1 and phase 2 stage A and appreciate some families may develop research fatigue. Families who already took part in phase 1 will be asked whether they would like to also take part in phase 2 stage A or prefer to opt-out.

### Phase 2 stage B

Results from phase 2 stage A will be reviewed for negative and positive cases. Families who have outlier findings, in terms of their answers to the validated questionnaires, will be invited to interview in phase 2 stage B, with the aim of reviewing and refining themes to develop additional theoretical explanation for the influence of psychosocial factors on transplant access and outcomes. The exact parameters to define outliers and sample size will be dependent on findings from phase 2 stage A.

## Outcomes measured

### Phase 1

Using a Topic Guide (online supplemental material 1), qualitative data will be collected in phase 1 to address research objectives 1–4 through semistructured, in-depth interviews undertaken by the investigator (JSK). This format was chosen over focus groups to enable participation of younger or less verbal children and avoid further inconveniencing families through multiple research appointments by having separate interviews for young children and focus groups for carers. The family interview format will be based on the child's decision to either interview with or separately from their carers, depending on which setting they find more comfortable. To ensure no participant is unfairly excluded from interviews due to English not being their first language, interpreters will be present to support their participation. The same principle of minimising communication barriers will be applied to younger children or young people who may prefer communicating through other creative outputs such as drawing or Talking Mats. The research goals for family interviews will include exploring what families feel their life is like now living with CKD-5, what good HR-QoL looks like and what they believe delays or enables how soon they receive a kidney transplant. Similarly, research goals for MDT member interviews will include exploring what the professionals think matters most to families whose child has CKD-5 when it comes to a good HR-QoL and what they, as professionals, believe impacts how soon children access a kidney transplant in terms of psychosocial factors.

The Topic Guide has been developed from the investigator's systematic literature review on the subject matter and in consultation with the Young Persons Advisory Group and the Research Partner Family.[29] To minimise any inconvenience in joining the study, all participants will be interviewed using their preferred modality (telephone, video-call or face-to-face consultations) in keeping with COVID-19 safety recommendations.[30] Interview time with young children will be kept to less than half an hour to minimise interview fatigue. All interviews will be audio or video recorded depending on participant preference.

Participants may become distressed as they reflect on their experiences due to the sensitive nature of some of the interview topics around their mental health or transplant delays. Therefore, the participant and JSK will agree on a 'stop signal' before commencing the interview for use should they feel uncomfortable. If the participant uses the 'stop signal', they will be offered a break. The participant and JSK will then discuss whether they would like to continue, reschedule or withdraw from the interview altogether. Once the interview recording stops, there will be an opportunity to discuss the participant's feelings and, if appropriate, they will be signposted to their local support services.

The acceptability of existing validated age-appropriate questionnaires that measure outcomes relevant to the preliminary themes will be discussed with the steering group. Potential validated questionnaires that capture the preliminary themes will be identified from the systematic literature review and a wider search of the literature. These questionnaires will be checked in terms of their psychometric properties such as internal consistency (Cronbach alpha of at least 0.7) and test–retest reliability and aspects such as availability of the measures, respondent type (parent, child or other respondent) and age range for which the questionnaire has been validated, to enable the most appropriate questionnaire to be chosen to measure each theme. The selected questionnaires will then be discussed with the steering group, considering the language of the questionnaire, acceptability and level of burden for the participant. If the list of questionnaires is too onerous for the participating family, a consensus will be reached with the steering group on which preliminary themes and therefore which questionnaires should be prioritised. Once the final list of questionnaires is agreed on, these will be submitted to the Health Research Authority for final approval.

### Phase 2 stage A

In phase 2 stage A, research objectives 5–6 will be addressed by measuring the following primary outcome variables: HR-QoL, psychosocial functioning and time taken to receive a kidney transplant since the date confirming CKD-5. To understand changes in psychosocial factors over time, their associations with health burden or short-term allograft deterioration must be accounted for, Therefore, we will measure the following secondary outcome: the child's estimated glomerular filtration rate (eGFR) over time. The eGFR will be calculated with their height and serum creatinine using the CkiD U25 formula.[31 32]

Questionnaires measuring HR-QoL and psychosocial functioning, selected from phase 1, will be distributed to a larger, national cohort of children with CKD-5 and their carers at their pretransplant baseline. Follow-up questionnaires will be sent post-transplant at 3, 6 and 12 months later or 12 months after their first questionnaire if they still have not received a kidney transplant in that time frame. These follow-up time points were chosen to reflect the initial period of post-transplant adaptation, which is comparable with similar studies in adults and children at the time of protocol-writing.[33–35] For children who have not received a transplant, a 12-month interval was advised by our steering group to avoid distress triggered by frequent questionnaires reminding them of their non-transplanted state. As families are more likely to participate if they can choose which questionnaire modality is most suited to their lifestyle, participant families will be offered either paper or online questionnaires.[36] Health morbidity and coexisting disease data, including their underlying primary kidney diagnosis, as described in table 1, will be mapped to each questionnaire time point.

An additional questionnaire (see online supplemental material 2) has been codesigned with the steering group to retrieve information on the participating family's

**Table 1** Coexisting health data to be collected and mapped against questionnaire time points

| Baseline of all participating CYP | CYP not transplanted yet at 12 months follow-up | CYP who are post-transplant |
|---|---|---|
| ► Ethnicity<br>► Primary renal diagnosis<br>► Other comorbidity diagnoses<br>► Modality of KRT if applicable<br>► Date confirming CKD-5<br>► Date of starting KRT if applicable<br>► Date of previous kidney transplant reaching CKD-5 if applicable<br>► Height (cm)<br>► Weight (kg)<br>► Serum creatinine (µmol/L)<br>► Number and type of medications<br>► Post code* | ► Modality of KRT if changed (if applicable)<br>► Date of starting new modality of KRT (if applicable)<br>► Height (cm)<br>► Weight (kg)<br>► Serum Creatinine (µmol/L)<br>► Number and type of medications | Only at 3 months post-transplant<br>► Type of transplant kidney (living related/living unrelated/deceased donor (after cardiac or brain death)<br>► Donor and recipient mismatch<br>► ABO blood group compatibility<br>► Cold ischaemic time<br>► Need for plasma exchange/immunoadsorption/ Rituximab/intravenous immunoglobulin<br>At all post-transplant intervals (3, 6, 12 and 24 months)<br>► Changes to comorbidities if applicable<br>► Change of KRT modality if applicable<br>► Date of KRT modality change if applicable<br>► Height (cm)<br>► Weight (kg)<br>► Serum creatinine (µmol/L)<br>► Number and type of medications |

*Post code data will be converted to the relevant national multiple indices of deprivation (eg, English Indices of Deprivation, Scottish Index of Multiple Deprivation).
CKD-5, stage 5 chronic kidney disease; CYP, children and young people; KRT, kidney replacement therapy.

demographic background. Data on medication burden will also be collected due to its' possible confounding effect HR-QoL as described in current literature.[37]

### Phase 2 stage B
The Interview Topic Guide for phase 2 stage B has been designed with flexibility since the interview will depend on the nature of the outlier findings from phase 2 stage A.

### Data analysis plan
#### Phase 1
To address research objectives 1–4, JSK will thematically analyse interview transcripts, following the approach of Braun and Clarke.[38] Deidentified transcripts will be managed using NVivo software.[39] For the purposes of qualitative rigour, JSK will maintain a reflexivity journal after interviews and throughout data analysis, to ensure potential researcher biases and insights are noted. Maintaining a reflexivity journal is a gold-standard practice in qualitative research, which increases the credibility and deepens the understanding of the findings by describing the context in which data were collected and analysed.[40 41] A sample of transcripts will be examined by JW and discussed with JSK. Throughout all stages of data collection and analysis, findings will be shared and discussed by JSK, JW and SM.

#### Phase 2 stage A
Participant demographics will be described using descriptive statistics. It will also be used to address research objective 4 by describing the prevalence of psychosocial factors.

Research objective 6 will be addressed through the following:

First, descriptive statistics will again be used to describe how participants' psychosocial functioning and HR-QoL

variables change before and after transplantation at each data collection time point.

Second, to describe the impact waiting to access a kidney transplant and receiving one has on psychosocial functioning and HR-QoL over time, repeated measure analysis of variance will be undertaken.

Third, the association between clinical, demographic and psychosocial factors with accessing a kidney transplant will be measured using logistic regression modelling and Kaplan-Meier statistics.

Fourth, the association of clinical, demographic and psychosocial factors with failing transplant allograft in short-term follow-up between 1 and 2 years post-transplantation will be assessed. Where the outcome variable is the child's eGFR, linear regression modelling will be used. Where the outcome variable defines a failing post-transplant kidney as reaching an eGFR equivalent to stage IV CKD (eGFR 15–29 mL/min/1.73 m$^2$), logistic regression modelling and Kaplan-Meier statistics will be used.

Separately, the level of agreement between child and carer responses will be assessed. Choice of statistical tests will depend on the type of data, for example, Cohen's kappa for binary data, weighted kappa for ordinal data or interclass correlation coefficients for continuous data.

Where relevant, all statistical analyses will be adjusted for (but not limited to) child's age, gender, ethnicity, socioeconomic status and KRT status at baseline.

#### Phase 2 stage B
Interview data will be analysed using thematic analysis underpinned by the same principles as in phase 1.

## Phase 3

An integrative analysis of data from phases 1 and 2 will be undertaken to address research objective 7 and create a conceptual model of how psychosocial factors influence transplant access and outcomes. The quantitative and qualitative data will be reviewed together to understand where they converge, complement or diverge from the other dataset or have findings that are novel. A final lay report will be cowritten with the steering group to ensure it is meaningful, relevant and accessible to families whose child has CKD.

## Ethics

This study protocol has been peer reviewed by the National Institute for Health Research Academy and has been approved under IRAS number 270493 by the Wales Research Ethics Committee 4 (ref: 20/WA/0285) and the Scotland A Research Ethics Committee (ref: 21/SS/0038).

Coercive pressure of joining the study will be minimised. Research participants will not be paid for participation. Participant information sheets will indicate that there will be no added benefit or disruption to their medical care.

Informed written consent will be obtained from all participants aged ≥16 years old and written assent from participants aged 5–15 years old. Consent will be obtained for all research activities including interviews, questionnaires and retrieving health information from national databases and hospital records.

Participant confidentiality will be upheld by fully adhering to the Data Protection Act. Participant identifiers will be handled with appropriate pseudonymisation and all data will be kept on General Data Protection Regulation (GDPR) compliant encrypted devices or stored on hard drives with restricted access with the relevant encryption and password protection. The only instance where confidentiality will be breached is if a participant discloses information that has direct implications for child or adult safeguarding. Potential participants will be made aware of this as part of informed consent. All interviews will be digitally recorded, transcribed verbatim and have identifiable data redacted. Audio files will be transcribed either by JSK or by a third-party interview transcription company (Take Note). To minimise data handling breaches, all engagement with Take Note will only be through their secure web platform. All video files will be transcribed only by JSK to remove third-party involvement in the deidentification process. Once deidentified, only quotes that cannot lead to participant identification will be used in reports. Care will be taken in reporting findings to ensure individuals cannot be identified by their role, diagnosis, gender, age or geographic locality.

## Dissemination

Research participants will be updated about the findings through newsletters. Lay summaries approved by the steering group will be disseminated across charity websites accessed by children and families affected by CKD. To ensure that professionals who work with families affected by CKD are being reached, findings will be disseminated widely in relevant peer-reviewed journals and at national and international conferences. Finally, if appropriate, a dissemination strategy will be cocreated between JSK and the steering group for other professionals (eg, teachers) who may encounter vulnerable families with CKD and need early referral for psychosocial intervention.

**Author affiliations**
[1]Paediatric Nephrology, Great Ormond Street Hospital for Children NHS Foundation Trust, London, UK
[2]NIHR Great Ormond Street Hospital Biomedical Research Centre, London, UK
[3]Centre for Outcomes and Experience Research in Children's Health, Illness and Disability, Great Ormond Street Hospital for Children NHS Foundation Trust, London, UK
[4]Population, Policy and Practice, UCL Great Ormond Street Institute of Child Health, UCL, London, UK
[5]UK Renal Registry, Bristol, UK
[6]Population Health Sciences, University of Bristol Medical School, Bristol, UK
[7]Non-communicable disease epidemiology, London School of Hygiene & Tropical Medicine, London, UK
[8]Statistics and Clinical Studies, NHS Blood and Transplant, Bristol, UK

**Acknowledgements** We would like to thank the Great Ormond Street Hospital Young Person's Advisory Group and members of our Research Partner Family Steering Group—Emma Beeden, Katy Beeden, Angela Watt and Heather Davis from KDARS (Kidney Disease and Renal Support) for kids, for their expertise, support and advice. We would also like to thank members of the National Kidney Federation and Kidney Care UK for their continued support of this study. Finally, we would like to thank the British Association for Paediatric Nephrology, whose support and contribution towards this study has been instrumental to its launch.

**Collaborators** British Association for Paediatric Nephrology

**Contributors** JSK wrote the study protocol and coordinated the entire manuscript. DR contributed towards sample size calculation and statistical methods of study protocol. LP contributed towards statistical analysis of protocol and advised regarding UK Renal Registry involvement. DN contributed towards statistical analysis of protocol and advised regarding UK Renal Registry involvement. MR contributed towards statistical analysis of protocol and advised regarding NHS Blood and Transplant involvement. JW supervised JSK, study protocol development with regular input to initial drafts and final manuscript approval. SM supervised JSK, study protocol development with regular input to initial drafts and final manuscript approval.

**Funding** This work is supported by the National Institute for Health Research Academy, as a Doctoral Fellowship, grant number NIHR300727.

**Competing interests** JSK is the National Institute for Health Research Fellowship grant recipient, which funds this study. LP reports grants from the National Institute for Health Research and Kidney Research UK. She is also the paediatric research lead for the UK Renal Registry.

**Patient and public involvement** Patients and/or the public were involved in the design, or conduct, or reporting, or dissemination plans of this research. Refer to the Methods section for further details.

**Patient consent for publication** Not applicable.

**Provenance and peer review** Not commissioned; externally peer reviewed.

**ORCID iDs**
Ji Soo Kim http://orcid.org/0000-0001-6090-1650
Jo Wray http://orcid.org/0000-0002-4769-1211
Stephen D Marks http://orcid.org/0000-0001-9850-8352

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
