## [Reviewer comments · BMJ Open]

ARTICLE DETAILS

TITLE (PROVISIONAL)	Protocol for a multi-centre prospective exploratory mixed-methods study investigating the modifiable psychosocial variables influencing access to and outcomes after kidney transplantation in children and young people in the UK
AUTHORS	Kim, Ji Soo; Wray, Jo; Ridout, Deborah; Plumb, Lucy; Nitsch, Dorothea; Robb, Matthew; Paediatric Nephrology (BAPN), British Association for; Marks, Dr. Stephen

VERSION 1 – REVIEW

REVIEWER	Malik, Shafi University Hospitals Coventry and Warwickshire NHS Trust
REVIEW RETURNED	31-Jan-2024

GENERAL COMMENTS	Thank you for asking me to review this manuscript. Authors have presented the study protocol for a mixed methods study of psychosocial variables and their association with access to transplantation and subsequent outcomes in the paediatric population. It is a well designed study, language is appropriate, the study would be done in phases with results of phase 1 informing the choice of questionnaire to be used in phase 2. However, I have below comments to make with a view to improving the manuscript. 1. In phase 1 qualitative interviews, have authors considered doing interviews in a group format rather than individual interviews as the interaction between participants can bring out important themes. Appreciate for certain individuals and young children this may not be suitable, but may be suitable for care providers and carer.2. Improving access to transplantation is vital, the ATTOM study has shown the barriers to such. It may be important to recruit patients from all referring centres and not just the transplant centres to address the research question comprehensively.3. Choice of questionnaire for phase 2 - would be useful to the reader to explain why the results of phase 1 are so critical to inform the choice of questionnaire to be used in phase 2.4. If some patients get transplanted how will the analysis take in to consideration and adjust for the change in psychosocial outcomes compared to the same participants previous readings5. Will the study include both suspended and active patients6. Suggest to consider including hospitalization episode as a variable
--

	7. 3 months post tx data capture - please include graft function as variable 8. Faith and cultural factors as significant barriers in some community groups, it would be an opportunity to study these in addition, please consider including a question around this in the qualitative interview. 9. Demographic questionnaire - add postcode and region for assessing deprivation index.
--	---

REVIEWER	Plevinsky, Jill The Children's Hospital of Philadelphia
REVIEW RETURNED	27-Feb-2024

GENERAL COMMENTS	Thank you for the opportunity to review this paper, which details a protocol for a study that investigates psychosocial factors that influence access to and outcomes after kidney transplantation in children and young people (CYP). This is a mentored review; Dr. Kara West, a postdoctoral fellow at Children's Hospital of Philadelphia led this review with my support and mentorship. The protocol itself was developed in collaboration with a Research Partner Family, and approved by an advisory group that consists of patients and families with relevant lived experiences. Further, the study solicits and incorporates perspectives and input from multiple key sources (e.g., multidisciplinary team members, patients, and their caregivers) as part of its design. In addition to these notable strengths, the protocol is well-written and utilizes a staged approach with a mixed-method prospective, longitudinal design. Findings have the potential to contribute meaningful information to our understanding of psychosocial factors related to access to and outcomes after kidney transplant in pediatric populations. Below I offer several comments for the authors to address which would strengthen the manuscript and protocol. Introduction  1. It would be helpful to clarify what age range is encapsulated by "children and young people" in the Introduction to contextualize the literature reviewed (it is clarified in the Population section currently). 2. "Psychosocial factors" encompass a wide range of constructs at both the individual-, family-, and broader community-level. I recognize that the factors examined in this study are protocol-driven and are not specifically known at the outset; however, it would be helpful if the authors could operationalize or describe the scope of factors they are interested in or considering under the umbrella of "psychosocial factors." 3. Although reportedly little research in the UK has investigated psychosocial factors related to access to kidney transplantation and outcomes in pediatric populations, there has been work in this area in other countries that warrants acknowledgement (including prospective studies). See below for sample publications:  a. Anand, A., Malik, T. H., Dunson, J., McDonald, M. F., Christmann, C. R., Galvan, N. T. N., ... & Rana, A. (2021). Factors associated with long-term graft survival in pediatric kidney transplant recipients. Pediatric Transplantation, 25(4), e13999. b. Freiburger, D., Kimball, B., Traum, A. Z., Berbert, L., O'Melia, L., Daly, K. P., ... & McKenna, K. D. (2023). Equity factors in pediatric transplant listing: Initial findings from a single center review. Pediatric Transplantation, 27(2), e14467.
---

c. Rea, K. E., West, K. B., Dorste, A., Christofferson, E. S., Lefkowitz, D., Mudd, E., Schneider, L., Smith, C., Triplett, K., & McKenna, K. (2023). A systematic review of social determinants of health in pediatric organ transplant outcomes. *Pediatric Transplantation*, 1-33. doi: 10.1111/petr.14418

Methods and Analysis

4. In the Population section, it states that CYP between 0-17 years of age will be invited to participate. At which lower age limit do you plan to directly involve the pediatric patient, as opposed to just their carer(s). Based on a statement at the end of the protocol in the Ethics section, it appears the lower limit is 5 years of age. It would be helpful to provide this information sooner, as well as mention considerations regarding young children's ability to complete validated questionnaires.

5. In Phase Two Stage B, families who have "negative cases," or outlier findings, on their responses to validated psychosocial questionnaires will be identified. I am wondering why the authors chose to limit it to negative outliers, as research objectives 3, 5, and 7 state their goal of identifying psychosocial challenges OR facilitators (or positive and negative psychosocial factors) in accessing a kidney transplant. Positive outliers may aid in the identification of factors that facilitate access or promote positive outcomes after transplantation (as indicated in research objective 5). In some cases, positive and negative factors may represent opposite sides of the same coin, although not necessarily for all.

6. In the Population section for Phase 2 Stage A, it states that the study team will aim to recruit every CYP with CKD-5 who meets the inclusion criteria. Would this include CYP and their carer(s) who may have participated in Phase 1 interviews? Please clarify.

7. In the Outcomes Measured section for Phase 2 Stage A, it states that questionnaires will be sent post-transplant at pre-specified intervals or at 12-months after their first questionnaire if they have not been transplanted. I have two questions:

a. First, given that participants recruited for this stage include pre-transplant CYP, would all participants initially complete surveys pre-transplant? Please clarify in the manuscript text when discussing administration of measures (it seems clearer in the flow diagram in the appendix).

b. Second, I am wondering why the decision was made to not administer questionnaires at the same intervals for participants who have and have not received a transplant over the course of the two-year study period. That would allow for an interesting comparison of psychosocial factors between those who received a transplant versus those who did not, or waited longer for transplant, which seems consistent with the study aim of understanding psychosocial factors related to access to and outcomes after kidney transplant.

8. Relatedly, given that participants from Phase 2 Stage A will be selected for Phase 2 Stage B, I am wondering if transplant status (e.g., active waitlist, not on waitlist, transplanted) would impact their ability to be recruited and/or participate in Stage B.

9. The authors nicely link specific research objectives within the Outcomes Measured section to draw a direct parallel. It would be helpful if they did the same in Data Analysis Plan within each phase.

Ethics

10. In the Ethics section it states that interpreters will be used to ensure that no participant is unfairly excluded. How will this be

	accomplished for Phase 2 when validated measures are used, as many may not have been translated and/or validated for use with specific languages? Minor comments 11. Consider using less abbreviations for increased readability. Many of the abbreviations utilized are not commonly recognizable and may cause confusion for readers. 12. Consider overall streamlining of the content presented; perhaps use of headings and subheadings would allow for a more digestible protocol. 13. Figure 1 is extremely small and difficult to read – please reformat it so text is readable and abbreviations are explained.
--	--

VERSION 1 – AUTHOR RESPONSE

Reviewer: 1

Dr. Shafi Malik, University Hospitals Coventry and Warwickshire NHS Trust

1. In phase 1 qualitative interviews, have authors considered doing interviews in a group format rather than individual interviews as the interaction between participants can bring out important themes. Appreciate for certain individuals and young children this may not be suitable but may be suitable for care providers and carer.

We have addressed this in the first paragraph of “Outcomes Measured: Phase One” in the Methods and Analysis section on page 11. As iterated in this paragraph, we have chosen the in-depth interview format over focus groups to enable participation of younger or less verbal children. We acknowledge that interaction between participants can generate important themes. However, arranging an interview appointment for the child followed by a separate focus group for carers could risk research fatigue in the carers, inconveniencing the family through multiple appointments and the additional burden of finding childcare whilst participating in a carers-only focus group. Furthermore, to run a focus group with participants from multiple centres is also challenging in terms of logistics and may result in a lower uptake.

Improving access to transplantation is vital, the ATTOM study has shown the barriers to such. It may be important to recruit patients from all referring centres and not just the transplant centres to address the research question comprehensively.

Thank you for this point, as inferred by our paragraph in “Population” in the Methods and Analysis section, we aim to include all 13 UK paediatric nephrology units, 10 of which are transplant centres and 3 are units that do not perform transplantation surgery but refer their paediatric patients to the other 10 transplant centres.

2. Choice of questionnaire for phase 2 - would be useful to the reader to explain why the results of phase 1 are so critical to inform the choice of questionnaire to be used in phase 2.

We have added to the “Study Design Overview” in our “Methods and Analysis” section this statement: “...Using the interviews to inform questionnaire selection will ensure questionnaires are relevant to, and resonate with, families...”

3. If some patients get transplanted how will the analysis take into consideration and adjust for the change in psychosocial outcomes compared to the same participants previous readings

The aim of this study is to examine changes in psychosocial outcomes over time and we would certainly be able to explore the difference between those who were transplanted and those who were not. The comparison between the pre-transplant and post-transplant psychosocial outcomes is an important aspect of this study’s analysis.

4. Will the study include both suspended and active patients?

Yes, this has been clarified in the second paragraph of “Population” in the Methods and Analysis section “...or on the waiting list for deceased donation (active or suspended)...”

5. Suggest considering including hospitalization episode as a variable.

Although hospitalization episodes would be informative as a variable illustrating the degree of morbidity experienced by the family, the practicalities of ensuring all relevant hospitalisation episodes are documented may be challenging as not all inpatient episodes would be experienced in the patient’s referring centre or transplant centre but at a local or out-of-area hospital.

6. 3 months post tx data capture - please include graft function as variable

Thank you for highlighting this, we have amended Table 1 to ensure it is clear we are collecting data pertaining to allograft function in month 3 as well as 6, 12 and so-forth.

7. Faith and cultural factors as significant barriers in some community groups, it would be an opportunity to study these in addition, please consider including a question around this in the qualitative interview.

Thank you for this suggestion. We will undertake semi-structured interviews with participants, so our questions are not limited to those suggested in our topic guide. We anticipate there will be opportunities to explore faith and cultural factors that may contribute to transplant access/outcome barriers or facilitators with our participants.

8. Demographic questionnaire - add postcode and region for assessing deprivation index.

Post code data will be collected via the UK Renal Registry. Collection of this data point has been added to Table 1.

Reviewer: 2

Dr. Jill Plevinsky, The Children's Hospital of Philadelphia

Introduction

1. It would be helpful to clarify what age range is encapsulated by “children and young people” in the Introduction to contextualize the literature reviewed (it is clarified in the Population section currently) We have clarified this in the opening sentence of the introduction with: “Around 1,000 children and young people (age 0-17 years) with Stage 5 chronic kidney disease...”

2. “Psychosocial factors” encompass a wide range of constructs at both the individual-, family-, and broader community-level. I recognize that the factors examined in this study are protocol-driven and are not specifically known at the outset; however, it would be helpful if the authors could operationalize or describe the scope of factors they are interested in or considering under the umbrella of “psychosocial factors.”

Thank you for appreciating the challenge whereby “psychosocial factors” encompasses a very broad range of constructs. This protocol has been designed to avoid researcher-driven assumptions of what these psychosocial factors may be. We have included this statement in the first paragraph of our “Aims and Objectives”:

“We anticipate these psychosocial factors to be broad at an individual, family and societal level and that they will encompass mental health and social determinants of health as defined by Marmot: ‘the conditions in which people are born, grow, live, work and age’ – we hypothesise...”

3. Although reportedly little research in the UK has investigated psychosocial factors related to access to kidney transplantation and outcomes in pediatric populations, there has been work in this area in other countries that warrants acknowledgement (including prospective studies). See below for sample publications:

a. Anand, A., Malik, T. H., Dunson, J., McDonald, M. F., Christmann, C. R., Galvan, N. T. N., ... & Rana, A. (2021). Factors associated with long-term graft survival in pediatric kidney transplant recipients. *Pediatric Transplantation*, 25(4), e13999.

b. Freiberger, D., Kimball, B., Traum, A. Z., Berbert, L., O'Melia, L., Daly, K. P., ... & McKenna, K. D. (2023). Equity factors in pediatric transplant listing: Initial findings from a single center review. *Pediatric Transplantation*, 27(2), e14467.

c. Rea, K. E., West, K. B., Dorste, A., Christofferson, E. S., Lefkowitz, D., Mudd, E., Schneider, L., Smith, C., Triplett, K., & McKenna, K. (2023). A systematic review of social determinants of health in pediatric organ transplant outcomes. *Pediatric Transplantation*, 1-33. doi: 10.1111/ptr.14418

We appreciate these publications being highlighted; we have now acknowledged these studies in the final paragraph of our "Introduction". They certainly add to current research in this field but as pointed out, there is a paucity of UK data and prospective longitudinal studies.

Methods and Analysis

4. In the Population section, it states that CYP between 0-17 years of age will be invited to participate. At which lower age limit do you plan to directly involve the pediatric patient, as opposed to just their carer(s). Based on a statement at the end of the protocol in the Ethics section, it appears the lower limit is 5 years of age. It would be helpful to provide this information sooner, as well as mentioned considerations regarding young children's ability to complete validated questionnaires.

We have now addressed this by mentioning that 5 years old will be the age at which children themselves will be invited to participate, in the second paragraph of "Population" in the "Methods and Analysis" section.

5. In Phase Two Stage B, families who have "negative cases," or outlier findings, on their responses to validated psychosocial questionnaires will be identified. I am wondering why the authors chose to limit it to negative outliers, as research objectives 3, 5, and 7 state their goal of identifying psychosocial challenges OR facilitators (or positive and negative psychosocial factors) in accessing a kidney transplant. Positive outliers may aid in the identification of factors that facilitate access or promote positive outcomes after transplantation (as indicated in research objective 5). In some cases, positive and negative factors may represent opposite sides of the same coin, although not necessarily for all.

Thank you for this suggestion, we have now included positive cases as well as negative cases.

6. In the Population section for Phase 2 Stage A, it states that the study team will aim to recruit every CYP with CKD-5 who meets the inclusion criteria. Would this include CYP and their carer(s) who may have participated in Phase 1 interviews? Please clarify.

As implied, there will be some overlap between families from Phase 1 and Phase 2 Stage A. We will offer families in Phase 1 the opportunity to opt out or take part in Phase 2 Stage A, as clarified in our final paragraph of "Phase Two Stage A Questionnaires" in "Population" in the "Methods and Analysis" section.

7. In the Outcomes Measured section for Phase 2 Stage A, it states that questionnaires will be sent post-transplant at pre-specified intervals or at 12-months after their first questionnaire if they have not been transplanted. I have two questions:

a. First, given that participants recruited for this stage include pre-transplant CYP, would all participants initially complete surveys pre-transplant? Please clarify in the manuscript text when discussing administration of measures (it seems clearer in the flow diagram in the appendix).

Thank you for highlighting this, we have made the following two adjustments to the manuscript:

i. Study Design Overview

"These interviews will inform which questionnaires will be distributed at baseline and follow-up to the wider cohort of children with CKD-5 and their carer(s) in phase two stage A."

ii. Second paragraph of "Phase Two Stage A" in the "Outcomes measured" in the "Methods and Analysis" section:

"Questionnaires measuring HR-QoL and psychosocial functioning, selected from phase one, will be distributed to a larger, national cohort of children with CKD-5 and their carers at their pre-transplant baseline."

b. Second, I am wondering why the decision was made to not administer questionnaires at the same intervals for participants who have and have not received a transplant over the course of the two-year study period. That would allow for an interesting comparison of psychosocial factors between those who received a transplant versus those who did not, or waited longer for transplant, which seems consistent with the study aim of understanding psychosocial factors related to access to and outcomes after kidney transplant.

Yes, we acknowledge the differences in intervals for non-transplanted and post-transplant families. This was raised with our research partner family in the beginnings of our research design, and it was felt that this may create mental stress for some families as they would be receiving questionnaires in quick succession reminding them of the fact that they had not yet received their transplant. We felt that it was important that we acknowledged their viewpoint and so designed our questionnaire administration time accordingly. We have added a statement alluding to this in the 2nd paragraph of Phase 2 Stage A segment in our 'Outcomes measured' section.

8. Relatedly, given that participants from Phase 2 Stage A will be selected for Phase 2 Stage B, I am wondering if transplant status (e.g., active waitlist, not on waitlist, transplanted) would impact their ability to be recruited and/or participate in Stage B.

We acknowledge that there may be issues recruiting participants to stage 2B but we do not think that transplant status per se is likely to be a reason. We do recognise that if the child is very sick either before or after transplant they or their family may not be able or willing to participate. It is also possible that families who are 'negative cases' rather than 'positive cases' may be less willing to be interviewed.

9. The authors nicely link specific research objectives within the Outcomes Measured section to draw a direct parallel. It would be helpful if they did the same in Data Analysis Plan within each phase. We have utilised this suggestion and adjusted our Data analysis plan to refer back to our research objectives.

Ethics

10. In the Ethics section it states that interpreters will be used to ensure that no participant is unfairly excluded. How will this be accomplished for Phase 2 when validated measures are used, as many may not have been translated and/or validated for use with specific languages?

We acknowledge the potential lack of validated translations of existing questionnaires may limit inclusion into Phase 2. We have amended the fourth paragraph of our Ethics section "Ideally similar principles would be applied to phase two questionnaires, but the availability of translated and validated questionnaires is a limitation." and added another bullet point to the Article Summary to acknowledge this limitation.

Minor comments

11. Consider using less abbreviations for increased readability. Many of the abbreviations utilized are not commonly recognizable and may cause confusion for readers.

Thank you for this suggestion – we have removed less common abbreviations as listed in the abbreviations list throughout the text for the reader's ease.

12. Consider overall streamlining of the content presented; perhaps use of headings and subheadings would allow for a more digestible protocol.

We have added additional headings: "Aims and Objectives" and "Conclusion". It may be a formatting issue but we have underlined all of our headings and subheadings to allow for easier identification in the main manuscript text.

13. Figure 1 is extremely small and difficult to read – please reformat it so text is readable and abbreviations are explained.

We have now adjusted the font for Figure 1.

We look forward to hearing back from you and thank you again for your time reviewing our manuscript.

VERSION 2 – REVIEW

REVIEWER	Malik, Shafi University Hospitals Coventry and Warwickshire NHS Trust
REVIEW RETURNED	08-Apr-2024

GENERAL COMMENTS	Authors have addressed questions and made recommended changes to the manuscript.
--

REVIEWER	Plevinsky, Jill The Children's Hospital of Philadelphia
REVIEW RETURNED	16-Apr-2024

GENERAL COMMENTS	Overall, I think the manuscript is significantly improved from the previous version and I appreciate the effort the authors put in to address each point raised in the original reviews. I do not have any additional suggestions at this time, and believe this study and its findings will make a meaningful contribution to the literature.
--

VERSION 2 – AUTHOR RESPONSE